# Tonsillectomy among children with low baseline acute throat infection consultation rates in UK general practices: a cohort study

Elizabeth Koshy,[1] Hilary Watt,[1] Vasa Curcin,[2] Alex Bottle,[1] Mike Sharland,[3] Sonia Saxena[1]

▶ Prepublication history and additional material is available. To view please visit the journal (http://dx.doi.org/10.1136/bmjopen-2014-006686).

[1]Department of Primary Care and Public Health, Imperial College London, London, UK
[2]Department of Primary Care and Public Health Sciences, Kings College London, London, UK
[3]Paediatric Infectious Diseases Research Group, St George's University London, London, UK

**Correspondence to**
Dr Elizabeth Koshy;
e.koshy@imperial.ac.uk

## ABSTRACT

**Objective:** To investigate the effectiveness of tonsillectomy in reducing acute throat infection (ATI) consultation rates over 6 years' follow-up among children with low baseline ATI consultation rates.

**Design:** Retrospective cohort study.

**Setting:** UK general practices from the Clinical Practice Research Datalink.

**Participants:** Children aged 4–15 years with ≤3 ATI consultations during the 3 years prior to 2001 (baseline). 450 children who underwent tonsillectomy (tonsillectomy group) and 13 442 other children with an ATI consultation (comparison group) in 2001.

**Main outcome measures:** Mean differences in ATI consultation rates over the first 3 years' and subsequent 3 years' follow-up compared with 3 years prior to 2001 (baseline); odds of ≥3 ATI consultations at the same time points.

**Results:** Among children in the tonsillectomy group, the 3-year mean ATI consultation rate decreased from 1.31 to 0.66 over the first 3 years' follow-up and further declined to 0.60 over the subsequent 3 years' follow-up period. Compared with children who had no operation, those who underwent tonsillectomy experienced a reduction in 3-year mean ATI consultations per child of 2.5 (95% CI 2.3 to 2.6, p<0.001) over the first 3 years' follow-up, but only 1.2 (95% CI 1.0 to 1.4, p<0.001) over the subsequent 3 years' follow-up compared with baseline, respectively. This equates to a mean reduction of 3.7 ATI consultations over a 6-year period and approximates to a mean annual reduction of 0.6 ATI consultations per child, per year, over 6 years' follow-up. Children who underwent tonsillectomy were also much less likely to experience ≥3 ATI consultations during the first 3 years' follow-up (adjusted OR=0.12, 95% CI 0.08 to 0.17) and the subsequent 3 years' follow-up (adjusted OR=0.24, 95% CI 0.14 to 0.41).

**Conclusions:** Among children with low baseline ATI rates, there was a statistically significant reduction in ATI consultation rates over 6 years' follow-up. However, the relatively modest clinical benefit needs to be weighed against the potential risks and complications associated with surgery.

### Strengths and limitations of this study

- This large cohort study is among the first to explore the longer term impact of tonsillectomy among children with low baseline acute throat infection (ATI) consultation rates in a general practice setting in the UK.
- We examined ATI consultation history over a 6-year follow-up period using routine, patient-level general practice data from a nationally representative population-based data set, which limited response bias.
- The main limitations relate to the diagnostic coding of ATIs by clinicians in general practice and that attendance for ATI consultations is influenced by the health-seeking behaviour of parents and the management of ATIs by general practitioners.

## INTRODUCTION

Tonsillectomy is one of the most common operations performed in children and the majority are performed for frequent acute tonsillitis episodes, for which there is evidence of benefit.[1–4] However, differentiating between acute tonsillitis, pharyngitis and other causes of acute throat infections (ATIs) is challenging in a general practice setting. Children who only have a record of low frequency ATI rates can undergo the operation, despite a lack of evidence for its clinical effectiveness in this group.[1 5–8] Therefore, in children with low baseline frequency of ATI, clarity is needed about the potential benefits of tonsillectomy over time. Randomised controlled trials (RCTs) have typically not looked beyond 2 or 3 years.[4 7–9] It is difficult to justify the costs, ethics and logistical problems of mounting new trials over an extended period of time to address this gap in the literature.

The natural history of ATI in children is to spontaneously resolve.[1 8] This can make it difficult to assess whether tonsillectomy has

been effective or whether the ATI frequency would have attenuated over time. Therefore, it is important to compare the prior and subsequent ATI history among children who undergo tonsillectomy with those who do not to evaluate its impact over an extended period of time. Information about the medium and longer term benefits of tonsillectomy on key outcomes including reduction in subsequent ATI is important for clinicians and to aid parental decision-making. About 98% of children are registered with a general practice in the UK and the vast majority of consultations occur in this setting, which enables real-time ATI consultations to be studied over time.[10] [11] Our aim was to investigate whether children with low baseline ATI consultation rates who underwent tonsillectomy experienced a significant reduction in subsequent ATI consultation rates compared with children who did not undergo surgery over 6 years' follow-up, using longitudinal data from the Clinical Practice Research Datalink (CPRD) database.

## MATERIALS AND METHODS
### Data source
We used data from the CPRD, which contains individual-level computerised medical records from over 600 general practices and covers data for about 8% of the UK population.[12] The data include demographic details, prescriptions and clinical events, which are entered as diagnostic Read codes by general practitioners (GPs) during the routine clinical management of patients. CPRD has been extensively used for research.[12]

### Study design, ATI definition and study population
We conducted a cohort study of children aged 4–15 years who had ≤3 ATI consultations at baseline. Children with a record of tonsillectomy or adenotonsillectomy between 1 January and 31 December 2001 formed the tonsillectomy group. The comparison group comprised of children with no history of tonsillectomy but a record of a sore throat consultation between these dates (see online supplementary figure S1). The date for the tonsillectomy record or sore throat consultation in 2001 represented the index date. The study period spanned 9 years in total, comprising 3 years prior to the index date (baseline) and 6 years subsequent to the index date. The period of follow-up for subsequent consultations was from the index date, which was the date of the first tonsillectomy record or sore throat consultation in 2001, to the date of censoring (the latest CPRD data collection, end of the study period of 6 years postindex date or the patient's transfer out of the practice, whichever came first). We studied ATI consultations prior to and subsequent to the index date.

We developed a list of Read codes (see online supplementary table S2) to define prior and subsequent ATI consultations, which encompassed sore throat symptoms and signs, tonsillitis, pharyngitis and also non-specific upper respiratory tract infection (URTI)

codes used by GPs to record a throat infection consultation. We searched for non-specific URTI in addition to sore throat consultations, as our previous study showed that the yield can be low by searching exclusively for sore throat consultation record codes.[5]

To enable us to account for prior history of ATI, as a marker of severity, we included children aged 4–15 years if they had ≥3 years' registration prior to the index date (baseline; see online supplementary figure S1). We focused our target population on children with low baseline ATIs (with a mild severity spectrum) by excluding those with >3 ATI consultations during the baseline period (1998–2001). Children with a history of snoring, hypertrophied tonsils, sleep apnoea, polysomnography or obstructive sleep apnoea were also excluded from both groups, as this is a different indication for tonsillectomy and our focus was to examine decision-making about tonsillectomy for ATI, not the sleep-disordered breathing (SDB) spectrum. Children aged <4 years were excluded, as preschool children are more commonly referred for the SDB spectrum than for recurrent ATI.[13] The final study population consisted of 450 children in the tonsillectomy group and 13 442 children in the comparison group (see online supplementary figure S1).

### Outcome measures
Our main outcome was the mean ATI consultation rate per child compared over three time frames across the 9-year study period (1998–2007): 3 years prior to the index date (1998–2001), which is used as a baseline comparator in the main analyses, 1–3 years' follow-up postindex date (2002–2004) and 4–6 years' follow-up postindex date (2005–2007; table 2). We also calculated the odds of ≥3 vs <3 ATI consultations per child during the same follow-up time frames.

### Potential risk factors
We examined age, sex, previous ATI consultations and antibiotic history prior to tonsillectomy. Prior ATI consultation frequency was a proxy measure for the severity of ATI infection preceding tonsillectomy. We also examined a number of comorbid conditions, which are potentially associated with increased risk of URTI and tonsillectomy: asthma, Down's syndrome, cardiac disorders including congenital heart defects, immunodeficiency, diabetes mellitus, as well as previous inhaled corticosteroid use.[3] [14] We studied practice-level deprivation using 'Index of Multiple Deprivation' scores based on the location of the general practice.

### Statistical analysis
We used Fisher's exact test to investigate the differences in the baseline categorical characteristics of children in both study groups and the unpaired t test to compare continuous variables. We calculated the mean (with 95% CIs) for ATI consultations per child for both groups at: baseline (1998–2001), 1–3 years postindex date (2002–2004) and 4–6 years postindex date (2005–2007), in

those with sufficient follow-up. We used the paired t test to compare the mean differences within each group, followed by the unpaired t test to compare the mean differences between the two groups. A two-sided p value of <0.05 was considered to be statistically significant.

Finally, we used logistic regression to investigate the outcomes of: ≥3 vs <3 ATI consultations, in 2002–2004 and 2005–2007 compared with the baseline in 1998–2001, after adjusting for potential confounders. This binary cut-off in ATI consultation frequency was selected to potentially approximate to at least one ATI consultation per year postindex date, respectively, in both time periods. We calculated crude ORs followed by adjusted OR, after controlling for the following potential confounders: age, sex, the comorbid conditions aforementioned including at least one prior corticosteroid inhaler prescription, prior ATI consultations, prior antibiotics, practice-level deprivation and clustering of general practices. We tested for an interaction between age and sex for tonsillectomy a priori based on the observations from previous studies.[2] [15] There were no missing data for the sociodemographic variables (age, sex and practice deprivation score). However, the completeness levels of data for the passive smoking and body mass index variables were very poor (<10%), so we excluded these variables and did not adjust for them in the logistic regression analysis. Analysis was performed using Stata/SE V.11.1.[16]

## RESULTS

For the tonsillectomy group, there were 450 children with at least 3 years of prior registration history and ≤3 ATI consultations who underwent tonsillectomy in 2001 (index date). Among these children, 92% (414/450) also had 3 years' follow-up and 80% (358/450) had 6 years' follow-up postindex date, respectively. For the comparison group, there were 13 442 children with at least 3 years of prior history and ≤3 ATI consultations. Among these children, 90% (12 053/13 442) also had 3 years' follow-up and 79% (10 593/13 442) had 6 years' follow-up postindex date, respectively.

Among the 450 children who underwent tonsillectomy with ≤3 ATI consultations at baseline, the highest percentage of tonsillectomies were performed on children aged 4–7 years (46%, 209/450) and 53% (237/450) were performed on girls.

Children aged 4–15 years in the tonsillectomy group had a higher mean ATI consultation rate (1.3 over 3 years) than those in the comparison group (0.4 over 3 years) at baseline, 1998–2001 (table 1). The mean number of baseline antibiotic prescriptions was also higher in the tonsillectomy group (3.7 and 2.1 prescriptions over the 3 years in the tonsillectomy and comparison groups, respectively, table 1).

In the tonsillectomy group, the 3-year mean ATI consultation rate among children aged 4–15 years decreased from 1.31 to 0.66 between 1998–2001 and 2002–2004,

and was 0.60 in 2005–2007 (table 2). This represented a difference in mean ATI consultation rate of −0.65 (95% CI −0.78 to −0.51) between 1998–2001 and 2002–2004, and then of −0.72 (95% CI −0.88 to −0.56) in 2005–2007 compared with the same baseline (1998–2001).

By contrast, in the comparison group, the 3-year mean ATI consultation rate increased from 0.44 to 2.25 between 1998–2001 and 2002–2004, and was 0.93 in 2005–2007 (table 2). This represented a difference in mean ATI consultation rate of +1.81 (95% CI 1.78 to 1.84) between 1998–2001 and 2002–2004, and then of +0.49 (95% CI 0.46 to 0.52) in 2005–2007 compared with 1998–2001.

Among children aged 4–15 years, comparing those who underwent tonsillectomy with those who did not, the unpaired t test showed a net reduction in 3-year mean ATI consultations per child of 2.46 (95% CI 2.29 to 2.63, p<0.001) over 1–3 years' follow-up and of 1.21 (95% CI 1.04 to 1.38, p<0.001) over 4–6 years' follow-up, compared with baseline. This represents a net reduction of 3.67 ATI consultations over 6 years' follow-up or approximates to an annual reduction in mean ATI consultations of 0.61 per child, per year, over 6 years' follow-up.

The crude OR for the effect of tonsillectomy on the outcome of ≥3 ATI consultations over 1–3 years' follow-up was 0.18 (95% CI 0.13 to 0.27) and the adjusted OR was 0.12 (95% CI 0.08 to 0.17). The crude OR for ≥3 ATI consultations over 4–6 years' follow-up was 0.36 (95% CI 0.22 to 0.59), while the adjusted OR was 0.24 (95% CI 0.14 to 0.41). The crude ORs were adjusted for age, sex, the comorbid conditions mentioned earlier, including at least one prior corticosteroid inhaler prescription, prior ATI consultations, prior antibiotics, practice-level deprivation and clustering of general practices. Prior ATI consultations, followed by prior antibiotics, were the strongest confounders of the association between tonsillectomy and the outcome of ATI consultations.

## DISCUSSION
### Main findings
Children aged 4–15 years with a history of low ATI consultation frequency at baseline (≤3 ATI consultations during 3 years) who underwent tonsillectomy experienced a net reduction in the mean ATI consultation rate of 3.67 over 6 years' follow-up, representing 2.46 (95% CI 2.29 to 2.63, p<0.001) over the first 3 years' and 1.21 (95% CI 1.04 to 1.38, p<0.001) per child over the subsequent 3 years' follow-up, compared with those who did not undergo surgery. This approximates to an annual reduction in the mean ATI consultation rate of 0.61 per child, per year, over 6 years' follow-up. Tonsillectomy was also associated with much lower odds of ≥3 ATI consultations throughout the 6-year follow-up period.

### Strengths and limitations of the study
This is among the first studies to explore the longer term impact of tonsillectomy in a general practice setting in the UK, covering 9 years. We analysed ATI

**Table 1** Characteristics of the study population at baseline (1998–2001) in the tonsillectomy and no tonsillectomy (comparison) groups

| | Tonsillectomy ≤3 ATI consultations (N=450) | No tonsillectomy ≤3 ATI consultations (N=13 442) | t Test |
|---|---|---|---|
| Mean (SD) age | 8.6 (3.4) | 9.3 (3.6) | **p<0.001** |
| Mean (SD) number of ATI consultations in 3 years prior to index date | 1.3 (1.1) | 0.4 (0.8) | **p<0.001** |
| Mean (SD) number of antibiotics in 3 years prior to index date | 3.7 (3.9) | 2.1 (2.7) | **p<0.001** |

| | n | Per cent | n | Per cent | $\chi^2$ test |
|---|---|---|---|---|---|
| Sex | | | | | |
| Girls | 237 | 53 | 7285 | 54 | p=0.52 |
| Boys | 213 | 47 | 6157 | 46 | |
| Deprivation quintile | | | | | |
| 1 (least deprived) | 105 | 23 | 2848 | 21 | p=0.10 |
| 5 (most deprived) | 103 | 23 | 3545 | 26 | |
| Comorbidities | | | | | |
| Asthma | 91 | 20 | 2302 | 17 | p=0.10 |
| Cardiac disease and defects | 2 | 0 | 11 | 0 | p=0.07 |
| Down's syndrome | 3 | 1 | 15 | 0 | p=0.02 |
| Immunodeficiency | 0 | 0 | 1 | 0 | p=1.00 |
| Diabetes | 6 | 1 | 239 | 2 | p=0.59 |
| ≥1 corticosteroid inhaler prescription | 120 | 27 | 3259 | 24 | p=0.24 |

p Values shown in bold where statistically significant.
ATI, acute throat infection.

consultation data for up to 6 years' follow-up in both groups and, to our knowledge, such a long follow-up period has not been studied before. We used routine individual-level general practice data from a nationally representative population-based data set, which limited response bias. The large number of comparisons from a wide population base increased the power of the study.

However, there are also a number of limitations to our study. First, our measure of ATI frequency relied on visits to the GP rather than self-reporting. Although this is an objective measure, we recognise it relies on health-seeking behaviour (HSB) and could, for example, underestimate the frequency of ATI episodes since not every child attends with every ATI episode. Second, although we endeavoured to exclude all children who may have undergone tonsillectomy for the SDB spectrum, we are unlikely to have captured everyone who had surgery for this indication. Third, the losses to follow-up could have created attrition bias, but this is difficult to minimise and this also affects experimental studies. The losses to follow-up were similar in the tonsillectomy and comparison groups over the 3-year and 6-year follow-up periods and were both under 20% at 6 years' follow-up. Fourth, there were higher baseline ATI consultation and antibiotic means in the tonsillectomy group. However, we adjusted for prior ATI consultations and antibiotics which made the two groups more

comparable. Finally, the large size of the study population with demographic and comorbidity variables, which were available from the CPRD data set, allowed us to adjust for a number of potential confounding factors. However, as with most retrospective cohort studies, we were unable to study all known and unknown potential confounding variables and thus there could be residual confounding, which we could not control for.

### Comparisons with other studies

A Dutch RCT among children aged 2–8 years who did not fulfil the Paradise criteria[4] for severe recurrent throat infection frequency found the effects of tonsillectomy were more pronounced in children who had 3–6 throat infections in the year prior to entering the trial than in those with ≤2 throat infections.[8] Among children in this latter group, the authors reported fewer throat infections (−0.21, 95% CI −0.06 to −0.36) and fewer sore throats (−0.60, 95% CI −0.30 to −0.90) per person year in the tonsillectomy group but concluded that these differences were not large enough to be clinically significant, although the median follow-up time was only 22 months.[8] Although ours was a cohort study, we similarly found that although tonsillectomy led to a statistically significant reduction in ATI consultation rates among children with low baseline ATI, this was not necessarily clinically significant.

**Table 2** Mean differences in acute throat infection (ATI) consultations per child prior to and subsequent to the index date in 2001

| | Mean number of ATI consultations and mean differences (with 95% CIs) comparing 3-year time intervals (1998–2001, 2002–2004 and 2005–2007) | | | | | | |
|---|---|---|---|---|---|---|---|
| n | 1998–2001 3 years prior to index | 2002–2004 1–3 years postindex | Mean difference between 1998–2001 and 2002–2004† | n | 1998–2001 3 years prior to index | 2005–2007 4–6 years postindex | Mean difference between 1998–2001 and 2005–2007† |
| | 3 years prior to index date and 1–3 years' follow-up | | | | 3 years prior to index date and 4–6 years' follow-up | | |
| **All children** | | | | | | | |
| No tonsillectomy | 12 053 | 0.44 | 2.25 | +1.81 (1.78 to 1.84)*** | 10 593 | 0.44 | 0.93 | +0.49 (0.46 to 0.52)*** |
| Tonsillectomy | 414 | 1.31 | 0.66 | −0.65 (−0.78 to −0.51)*** | 358 | 1.32 | 0.6 | −0.72 (−0.88 to −0.56)*** |
| **Girls 4–7 years** | | | | | | | |
| No tonsillectomy | 2375 | 0.7 | 2.43 | +1.74 (1.66 to 1.81)*** | 2092 | 0.7 | 1.04 | +0.35 (0.27 to 0.42)*** |
| Tonsillectomy | 78 | 1.32 | 0.76 | −0.56 (−0.94 to −0.19)** | 71 | 1.34 | 0.65 | −0.69 (−1.08 to −0.31)*** |
| **Boys 4–7 years** | | | | | | | |
| No tonsillectomy | 2242 | 0.7 | 2.29 | +1.59 (1.51 to 1.67)*** | 1989 | 0.69 | 0.76 | +0.06 (0.00 to 0.13) |
| Tonsillectomy | 110 | 1.32 | 0.83 | −0.49 (−0.81 to −0.18)** | 92 | 1.32 | 0.55 | −0.76 (−1.02 to −0.50)*** |
| **Girls 8–11 years** | | | | | | | |
| No tonsillectomy | 1944 | 0.35 | 2.23 | +1.87 (1.80 to 1.95)*** | 1744 | 0.35 | 1.08 | +0.73 (0.65 to 0.81)*** |
| Tonsillectomy | 71 | 1.43 | 0.73 | −0.70 (−0.99 to −0.42)*** | 65 | 1.43 | 0.92 | −0.51 (−1.00 to −0.01)* |
| **Boys 8–11 years** | | | | | | | |
| No tonsillectomy | 1526 | 0.31 | 2.04 | +1.73 (1.65 to 1.80)*** | 1395 | 0.31 | 0.71 | +0.40 (0.33 to 0.47)*** |
| Tonsillectomy | 59 | 1.17 | 0.34 | −0.83 (−1.16 to −0.51)*** | 49 | 1.22 | 0.39 | −0.84 (−1.23 to −0.45)*** |
| **Girls 12–15 years** | | | | | | | |
| No tonsillectomy | 2206 | 0.23 | 2.45 | +2.22 (2.14 to 2.30)*** | 1866 | 0.23 | 1.2 | +0.97 (0.89 to 1.04)*** |
| Tonsillectomy | 66 | 1.29 | 0.62 | −0.67 (−0.95 to −0.38)*** | 55 | 1.22 | 0.62 | −0.60 (−0.96 to −0.24)** |
| **Boys 12–15 years** | | | | | | | |
| No tonsillectomy | 1760 | 0.2 | 1.91 | +1.70 (1.64 to 1.77)*** | 1507 | 0.21 | 0.68 | +0.47 (0.41 to 0.53)*** |
| Tonsillectomy | 30 | 1.27 | 0.37 | −0.90 (−1.28 to −0.52)*** | 26 | 1.42 | 0.23 | −1.19 (−1.69 to −0.69)*** |

*p<0.05.
**p<0.01.
***p<0.001.
†Paired t test.

A UK study which analysed CPRD data for annual sore throat consultation rates found that the rates per 1000 registered patients declined by 50% between 1995 and 2000 among children aged 5–16 years.[17] The authors reported sore throat consultation rates in 2000 of 59 and 76 per 1000 registered patients among children aged 5–10 and 11–16 years, respectively. However, that study did not investigate the annual consultation rates for an individual child. The authors proposed that the overall decline in consultation rates may reflect that patients were increasingly self-managing minor illnesses.[17]

We found the natural history among children of all ages who did not have tonsillectomy was for a gradual increase in the number of ATI consultations. This is contrary to previous studies, which reported that the natural history of sore throats is to attenuate over time.[18–20] However, our study was based in a community setting and focused on a population of children with low baseline ATI who may have emerging natural history whereas other studies have selected children with higher baseline ATI from the outset. If we had followed children up for longer we may have observed a decline over time.

### Clinical implications of the study

Overall, children aged 4–15 years, who underwent tonsillectomy compared with those who did not, experienced an 88% and 76% reduced risk of having ≥3 ATI consultations over 1–3 and 4–6 years' follow-up, respectively. However, the observed decline in ATI consultations might also, in part, be attributed to adenotonsillectomy operations reducing rhinosinusitis or otitis media infections, which were coded as URTIs.[21–23]

There was no evidence of an association between tonsillectomy and deprivation level, which is in contrast to some recent studies and reports.[24–27] However, we only had practice-level deprivation data and not individual-level data, which could account for the absence of an association.

Children in our study who underwent tonsillectomy in 2001 had ≤3 ATI consultations documented in general practice during the 3 years prior to their operation. This is markedly less than the minimum threshold recommended by national guidelines, reviews and the Paradise criteria for frequency of episodes prior to tonsillectomy.[1 4 20 28 29] These findings may reflect lower thresholds being followed or perhaps represent suboptimal documentation of sore throat consultations prior to tonsillectomy in primary care settings. Although we assumed that parents would have consulted if the ATI was severe, the low consultation rate could reflect HSB, whereby some parents may not present with ATI episodes if, for example, antibiotics were not issued at prior consultations. However, this depends on the individual GP's attitudes and threshold for prescribing antibiotics for ATIs.

Other possible explanations for these low ATI consultation rates are that we did not have access to the 'free text' within individual medical consultation records, which may contain additional details about ATI episodes. Children may have attended with concurrent signs or symptoms and so other diagnostic codes, such as otitis media or specific viral URTIs, may have been recorded with details of the sore throat symptoms included within the 'free text'. As previously mentioned, overall annual consultation rates for sore throat declined by 50% among children aged 5–16 years between 1995 and 2000 in the UK, which may further account for our lower ATI consultation rates at baseline.[17] We did not have consultations from other primary care settings, direct accident and emergency department attendances or hospital admissions, which have been increasing over recent years and may not be well documented within general practice records.[30] Therefore, we interpret our findings with the caveat that there could be other possible explanations for the low documentation rate of ATI consultations. However, the strength of our study is that we are consistent in comparing the ATI consultation frequency prior to and subsequent to the index date for each child in a particular general practice, which is where the vast majority of consultations with children occur.[10] We recommend future studies should examine individual consultations in greater detail, as well as attendances in community and hospital emergency settings, to attempt to capture ATI consultations within different healthcare settings.

Our findings suggest that there is limited clinical benefit in performing tonsillectomy for children with low baseline ATI consultation rates. This reinforces existing guidance that only children with severe throat infection disease at baseline should be referred for tonsillectomy. Previous research has suggested that the operation is beneficial in the short term for those with more serious ATI morbidity at baseline, although its benefit in the longer term is still not clear and this still needs to be investigated. Tonsillectomy is a major operation so it is important that surgery confers significant individual and societal benefits, including fewer days off work by parents affected by sick children.

Our study also highlights the importance of diligent documentation of ATI consultations within general practice. There is a need to validate diagnostic screening tools among children, such as the Centor criteria and McIsaac scores, for group A β-haemolytic streptococcal throat infections. This may help towards more accurate diagnosis, recording and more appropriate management of ATI consultations in general practice.

### CONCLUSION

Our study found a statistically significant reduction in ATI consultation rates among children with low baseline ATI consultation rates who underwent tonsillectomy compared with those who did not. However, for this subgroup of children, the relatively modest reduction in ATI consultation rates in clinical terms may be outweighed by the potential risks related to the operation. The long-term clinical effectiveness of tonsillectomy for children, across the ATI frequency spectrum, still needs to be evaluated. This needs to be undertaken in conjunction with updated cost-effectiveness models,

stratified by age, sex and baseline ATI rates, so that clinicians can risk profile children to determine which subgroups of children are most likely to benefit from surgery.

**Acknowledgements** Data were obtained from the Clinical Practice Research Datalink, CPRD (formerly known as the General Practice Research Database). The CPRD is jointly funded by the NHS National Institute for Health Research (NIHR) and the Medicines and Healthcare products Regulatory Agency (MHRA).

**Contributors** EK, SS and MS conceived the study. EK, SS, HW and AB designed the study. VC extracted the data. EK analysed the data. HW provided statistical advice. EK wrote the first draft of the manuscript. All authors contributed to the interpretation of the findings and revised the paper critically for intellectual content. All authors have seen and approved this version of the manuscript. EK and SS are the guarantors of this study.

**Funding** EK is funded by a National Institute for Health Research Doctoral Research Fellowship (2009-02-78). SS is funded by a National Institute for Health Research Career Development Fellowship (NIHR CDF-2011-04-048). HW is funded by NIHR Research Design Service. This article presents independent research funded by the National Institute for Health Research (NIHR). The views expressed are those of the authors and not necessarily those of the NHS, the NIHR or the Department of Health. The funders had no role in study design, data collection and analysis, decision to publish, or preparation of the manuscript.

**Competing interests** None.

**Ethics approval** The study was approved by the MHRA Independent Scientific Advisory Committee on Database Research.

**Provenance and peer review** Not commissioned; externally peer reviewed.

**Data sharing statement** No additional data are available.

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
