## [Reviewer comments · BMJ Open]

ARTICLE DETAILS

TITLE (PROVISIONAL)	Tonsillectomy among children with low baseline acute throat infection consultation rates in UK general practices: a cohort study
AUTHORS	Koshy, Elizabeth; Watt, Hilary; Curcin, Vasa; Bottle, Alex; sharland, mike; Saxena, Sonia

VERSION 1 - REVIEW

REVIEWER	Alex Toh United Kingdom
REVIEW RETURNED	08-Oct-2014

GENERAL COMMENTS	This is a valid study with clear objective. As the author concluded, the study shows statistically significant result but perhaps the improvement is not significant clinically. Perhaps it would be helpful to the readers if the author can suggest how the outcome of this study will affect/ improve current practice on managing acute throat infections in children?
--

REVIEWER	James Barraclough FRCS ENT Fellow Dunedin Hospital New Zealand
REVIEW RETURNED	12-Oct-2014

GENERAL COMMENTS	This retrospective cohort study has a clear question and I think the results and discussion manage to answer what is proposed. Most of the potential confounders have been addressed in the methodology and/or discussion. I have a few minor points to add. 1. I was amazed that so many children with low rates of sore throat attendance underwent tonsillectomy. This goes against the SIGN guidelines and a number of studies have shown that adherence to the guidelines is reasonable in the UK. This would lead to the conclusion that there is likely to be a disparity between the actual episodes reported to an ENT surgeon and the document episodes on the database. This study relies on the documentation of episodes as the primary measure. The authors do discuss this observation in that GPs may not be documenting the episodes accurately and many episodes may be seen in ED or not attend at all but this explanation has the consequence of diluting the accuracy of the data that is used in this study and hence the conclusions. Clearly this would be the case for both tonsillectomy and non tonsillectomy groups but this simple observation (high rate of operations in a seemingly non-indicated group) could be a quite a big flaw.
--

	2. It is worth mentioning the Centor criteria for diagnosis of tonsillitis as this is a reasonably well recognised way of diagnosing the condition. 3. line 35 - attendance for acute throat infection consultations is influenced 4. final sentence of conclusion - this needs to be... 5. It might be clearer to say that patients had 1-3 consultations rather than less than or equal to 3 as this would include patients who had no consultations (sorry to be pedantic) 6. A strength of the study that is not highlighted is that the results tell us that the influence of tonsillectomy for reporting of acute sore throats over time seems to last at least 6 years, certainly compared with the non-operated group who were worse even by this time period. Not many studies with a large population are available to tell us that. 7. The SIGN guidelines 117 are available fully online (see reference 23)
--	---

VERSION 1 – AUTHOR RESPONSE

Reviewer: 1

Comments to the Author

This is a valid study with clear objective. As the author concluded, the study shows statistically significant result but perhaps the improvement is not significant clinically. Perhaps it would be helpful to the readers if the author can suggest how the outcome of this study will affect/ improve current practice on managing acute throat infections in children?

Thank you for raising this point. We have addressed this point as follows:

“Our findings suggest that there is limited clinical benefit in performing tonsillectomy for children with low baseline ATI consultation rates. This reinforces existing guidance that only children with severe throat infection disease at baseline should be referred for tonsillectomy.

“.....Our study also highlights the importance of diligent documentation of ATI consultations within general practice. There is a need to carefully validate diagnostic screening tools among children, such as the Centor and McIsaac scores, for group A β haemolytic streptococcal throat infections. This may help towards more accurate diagnosis, recording and more appropriate management of ATI consultations in general practice.”

Reviewer: 2

Comments to the Author

This retrospective cohort study has a clear question and I think the results and discussion manage to answer what is proposed. Most of the potential confounders have been addressed in the methodology and/or discussion. I have a few minor points to add.

1. I was amazed that so many children with low rates of sore throat attendance underwent tonsillectomy. This goes against the SIGN guidelines and a number of studies have shown that adherence to the guidelines is reasonable in the UK. This would lead to the conclusion that there is

likely to be a disparity between the actual episodes reported to an ENT surgeon and the document episodes on the database. This study relies on the documentation of episodes as the primary measure. The authors do discuss this observation in that GPs may not be documenting the episodes accurately and many episodes may be seen in ED or not attend at all but this explanation has the consequence of diluting the accuracy of the data that is used in this study and hence the conclusions. Clearly this would be the case for both tonsillectomy and non tonsillectomy groups but this simple observation (high rate of operations in a seemingly non-indicated group) could be a quite a big flaw.

As Reviewer 2 highlights, we discuss the low rates of acute throat infection (ATI) consultations prior to tonsillectomy within our Discussion section and provide possible explanations for this observation. We have added more detail to highlight this further and we have added a sentence to reflect more cautious interpretation given the possible alternative reasons. However, we have applied a consistent approach by analysing ATI consultation frequency for each child before and after the index date within a particular general practice.

We have expanded this section with more detail:

“.....Children may have attended with concurrent signs or symptoms and so other diagnostic codes, such as otitis media or specific viral URTIs, may have been recorded with details of the sore throat symptoms included within the ‘free text’. We did not have consultations from other primary care settings, direct Accident and Emergency department attendances or hospital admissions, which have been increasing over recent years and may not be well-documented within general practice records. Therefore, we interpret our findings with the caveat that there could be other possible explanations for the low documentation rate of ATI consultations. However, the strength of our study is that we are consistent in comparing the ATI consultation frequency prior to and subsequent to the index date for every child in a particular general practice, which is where the vast majority of consultations with children occur.[9] We recommend future studies should examine individual consultations in greater detail, as well as attendances in community and hospital emergency settings, to attempt to capture more ATI consultations within different healthcare settings.”

We have also added the following section, which highlights a decline in overall sore throat consultation rates among children in the UK between 1995 and 2000:

“A UK study which analysed CPRD data for annual sore throat consultation rates found that the rates per 1000 registered patients declined by 50% between 1995 and 2000 among children aged 5-16 years. [1] The authors reported sore throat consultation rates in 2000 of 59 and 76 per 1000 registered patients among children aged 5-10 and 11-16 years, respectively. However, that study did not investigate the annual consultation rates for an individual child and did not focus on children with mild ATI disease. The authors suggested the overall decline in consultation rates may reflect that patients were increasingly self-managing minor illnesses. [1]

“..... As previously mentioned, overall annual consultation rates for sore throat declined by 50% among children aged 5-16 years between 1995 and 2000 in the UK, which may further account for the lower ATI consultation rates at baseline.[1]”

2. It is worth mentioning the Centor criteria for diagnosis of tonsillitis as this is a reasonably well recognised way of diagnosing the condition.

Thank you for this. We have added the following sentences to the Discussion:

“.....There is a need to carefully validate diagnostic screening tools among children, such as the Centor and McIsaac scores, for group A β haemolytic streptococcal throat infections. This may help

towards more accurate diagnosis, recording and more appropriate management of ATI consultations in general practice.”

3. line 35 - attendance for acute throat infection consultations is influenced

Thank you for noting this error. We have corrected it within the manuscript.

4. final sentence of conclusion - this needs to be...

Thank you for noticing the missing word in this sentence. We have corrected it within the manuscript.

5. It might be clearer to say that patients had 1-3 consultations rather than less than or equal to 3 as this would include patients who had no consultations (sorry to be pedantic)

There were some children who did not have a record of ATI consultations during the baseline period. The two primary indications for tonsillectomy are throat infections and obstructive sleep apnoea syndrome (OSAS). Throat infections are responsible for the majority of cases (among children aged 5 to <16 years, 89% of tonsillectomies were performed for throat infections, according to the Royal College of Surgeon's prospective audit of tonsillectomy in 2003/4). [2] We made every effort to exclude children who underwent tonsillectomy for OSAS, by excluding young children (aged <4 years), who are the most likely to undergo tonsillectomy for OSAS and by excluding children with a diagnostic code suggestive of OSAS in their medical records. Therefore, the most likely indication for tonsillectomy among the children we studied would have been for throat infections, as tonsillectomy is not indicated for any other type of recurrent upper respiratory tract infection. Hence we made the assumption that the children included in our tonsillectomy group were operated on for throat infections, as opposed to OSAS. Van Staaji et al also reported on the impact of tonsillectomy among children with 0-2 sore throat or URTI episodes at baseline on subsequent episodes, although they only followed-up children for a median of 22 months[3].

6. A strength of the study that is not highlighted is that the results tell us that the influence of tonsillectomy for reporting of acute sore throats over time seems to last at least 6 years, certainly compared with the non-operated group who were worse even by this time period. Not many studies with a large population are available to tell us that.

Thank you very much for highlighting this strength of our study. We have subsequently added the following to the Discussion:

“We analysed ATI consultation data for up to six years' follow-up in both groups and, to our knowledge, such a long follow-up period has not been studied before.”

7. The SIGN guidelines 117 are available fully online (see reference 23)

Thank you for this point. We have modified the references as follows:

Reference 23 (SIGN 34 guidelines 1999): Although SIGN 117 (the 2010 updated version of the guidelines) is available within the SIGN guidelines website the previous guidelines (SIGN 34, published in 1999) are no longer directly available from the SIGN website. However, we are now providing a URL link within reference 23, to access SIGN 34 guidelines from another online source.

Reference 1 (SIGN 117 guidelines 2010): We have also now added the URL for SIGN 117 guidelines within reference 1, so that it is easily accessible for the reader.

1. Ashworth M, Charlton J, Latinovic R, et al. Age-related changes in consultations and antibiotic prescribing for acute respiratory infections, 1995-2000. Data from the UK General Practice Research Database. *Journal of clinical pharmacy and therapeutics*. 2006 Oct;31(5):461-7.
2. Royal College of Surgeons of England (2005) National Prospective Tonsillectomy Audit final report of an audit carried out in England and Northern Ireland between July 2003 and September 2004.
3. van Staaïj BK, van den Akker EH, Rovers MM, et al. Effectiveness of adenotonsillectomy in children with mild symptoms of throat infections or adenotonsillar hypertrophy: open, randomised controlled trial. *BMJ*. 2004 Sep 18;329(7467):651.